# Association between the intake of dietary n3 and n6 fatty acids and stroke in US adults: A cross-sectional study of NHANES 2007–2018

**Mingya Luan[1], Jia Wang[2], Kun Liang[1], Bo Li[3], Kewei Liu[1]***

**1** General Practice Department of Medicine, 960th Hospital People's Liberation Army of China Joint Logist Support Force, Jinan, China, **2** Public Health Department, Weihai Maternal and Child Health Hospital, the Affiliated Weihai Second Municipal Hospital of Qingdao University, Weihai, China, **3** Zhangcun Town Health Center in Huancui District, Weihai, China

* liukewei1971@126.com

## Abstract

### Background

The association between the intake of dietary n3 and n6 fatty acids and the risk of stroke is subject to debate. The primary objective of the present research was to establish the correlation in a large sample of American adults.

### Methods

Using data from the National Health and Nutrition Examination Survey (NHANES) between 2007 and 2018, the association of the intake of dietary n3 and n6 fatty acids with stroke events was analyzed in a sample of 29,459 adults. The intake of n3 and n6 fatty acids intake was assessed though two 24-h dietary recalls. Stroke outcomes were identified based on the responses provided in self-reported questionnaire. Logistic regression was fitted to evaluate the correlation of dietary n3, n6 fatty acids intake with stroke events.

### Results

Subjects in the highest tertile (T3) of dietary n3 (OR: 0.67, 95% CI: 0.49–0.93), n6 (OR: 0.65, 95% CI: 0.45–0.95) fatty acids intake were found to have obviously lower risk of stroke compared to those in the lowest tertile (T1), but the n6:n3 ratio was not found to be associated with a stroke event. Results from stratified analysis demonstrated that dietary n3 fatty acids had an inverse correlation of stroke in both male and female, but dietary n6 fatty acids only had this correlation in male. Moreover, findings were made that the interaction was significant in terms of age in the subgroup analysis, and the negative relationship between the intake of dietary n3 and n6 fatty acids and stroke event were particularly pronounced among participants aged ≥60.

### Conclusions

The present results suggested that increased dietary n3, n6 fatty acids intake correlated with a lower risk of stroke.

**Data Availability Statement:** All relevant data are within the paper and its Supporting Information files.

**Funding:** The author(s) received no specific funding for this work.

**Competing interests:** The authors have declared that no competing interests exist.

## Introduction

After ischemic heart disease, stroke currently stands as the second leading cause of death [1, 2]. Stroke is also one of the leading global causes of long-term disability. Even for individuals who survive a stroke, their quality of life is often diminished, and they necessitate significant rehabilitation expenditures [3]. In 2019, there were reported to be 12.2 million cases of stroke, resulting in 6.55 million stroke-related deaths and the loss of 143 million disability-adjusted life-years due to stroke [4].

Dietary modification is an easy and low-cost way to prevent stroke [5], and numerous dietary factors, including protein [6, 7], fiber [8], dietary calcium [9], dietary selenium [10], nuts [11], dietary flavonoid [12], and dietary copper [13] are known to be associated with the risk of stroke. Evidence has demonstrated that n3 and n6 fatty acids can increase biomembrane fluidity [14], diminish the formation of free radicals [15], and inhibit apoptotic pathways. There have also been reports suggesting that a deficiency in the dietary intake of n3 and n6 fatty acids is associated with several chronic diseases, including hypertension [16, 17], diabetes [18, 19], and obesity [20, 21], all of which are recognized risk factors for stroke [22]. Thus, a deficiency in n3 and n6 fatty acids may have a connection with the development of stroke.

Several studies have explored the correlation of long-chain n3 fatty acids, obtained primarily from marine foods, with stroke risk; however, the results have been inconsistent [23–30]. A protective role of another plant-derived n3 fatty acids, alpha-linolenic acid, against stroke has also been found [31]. However, there is a scarcity of studies that have specifically investigated the relationship between the dietary intake of total n3 and n6 fatty acids and the risk of stroke. Herein, in the present study, a substantial sample from the National Health and Nutrition Examination Survey (NHANES) spanning the years 2007 to 2018 was employed to assess the association between dietary intake of n3 and n6 fatty acids and the risk of stroke.

## Methods

### Study population

Data were obtained from the NHANES database, a nationally representative study with the aim of assessing the health and nutritional status of Americans. Using a complex, stratified and multistage sample design, NHANES has been recruiting subjects since 1999 and publishing data every two years [32, 33]. The Ethics Review Committee of the National Center for Health Statistics approved the protocol and written informed consent was obtained from the participants [34].

In the present study, data from six investigation waves (2007–2018) were utilized, involving a total of 59,842 respondents. However, 30,383 subjects were excluded from the analysis due to the following exclusion criteria: age < 18 years (n = 23262), missing stroke data (n = 3256), missing dietary data (n = 2886), pregnant or lactating women(n = 469), inappropriate energy intake (n = 510). Ultimately, 29,459 subjects were included in the analysis (Fig 1 and S1 Table).

### Stroke outcomes

Questions from the Medical Conditions Questionnaire were used to identify the conditions of a stroke event. In the self-reported questionnaires, individuals were asked to answer the question that "Has a doctor or other health professional ever told you that you have had a stroke." If they responded affirmatively, they were classified as having experienced a stroke as an adverse outcome. Notably, this method of measuring self-reported stroke through questionnaires has demonstrated reliability and has been utilized in numerous prior research articles [13, 35, 36].

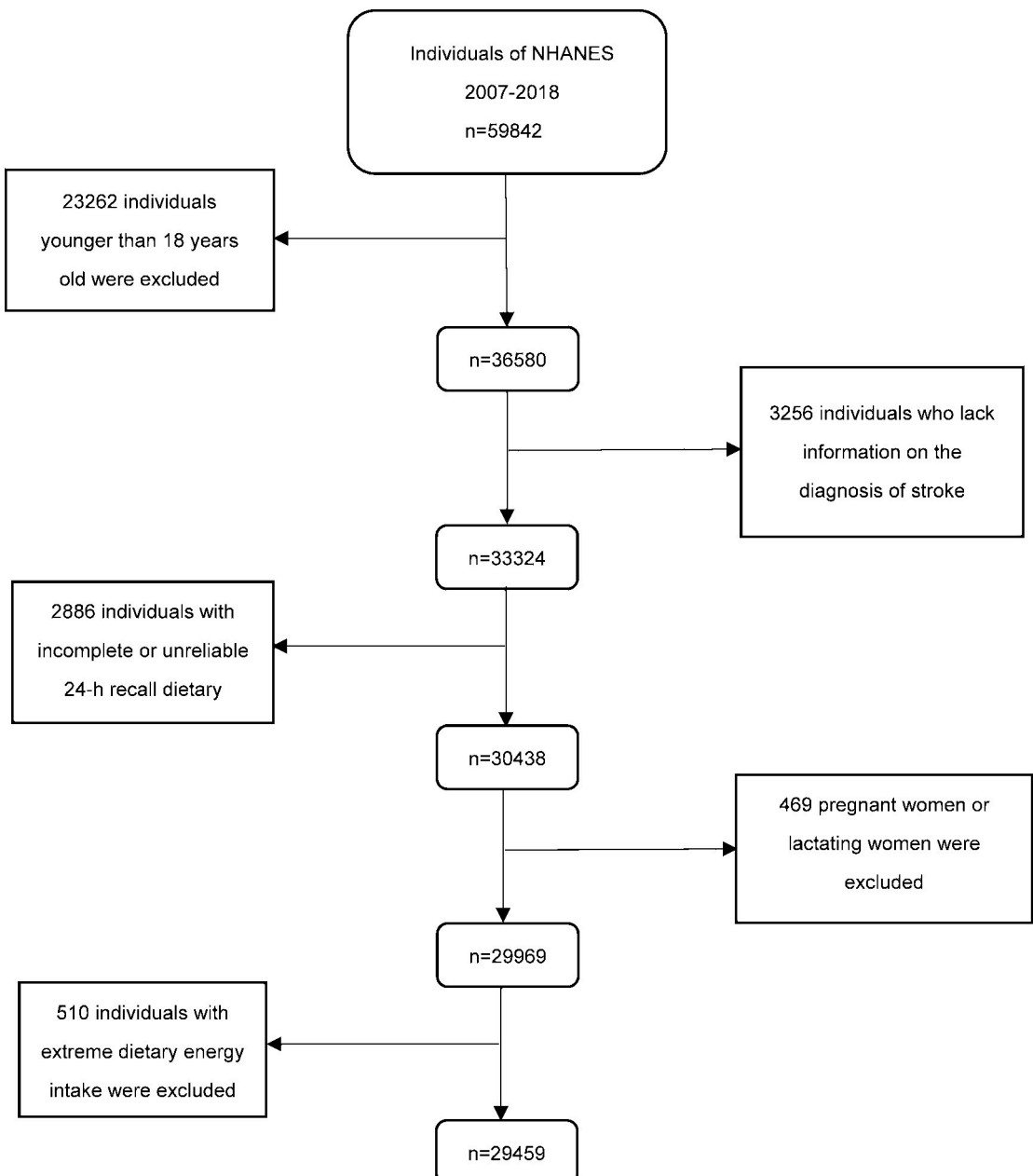

**Fig 1. Flow chart of the screening process for the selection of eligible participants.** NHANES: National Health and Nutrition Examination Survey.

## Assessment of the intake of dietary n3 and n6 fatty acids

The numerical value of the intake of dietary n3 and n6 fatty acids was obtained in the NHANES by means of two 24-h recall interviews. One was performed in the Mobile Examination Center (MEC), and the other one was conducted via telephone 3–10 days later. Given that linolenic acids primarily comprise "alpha-linolenic acids (n3)" with only a small quantity of "gamma-linolenic acids (n6)," and the NHANES database did not distinctly categorize linolenic acids, it was reasonable to treat them as n3 fatty acids in this study. Finally, the category of n3 fatty acids encompassed various fatty acids, including linolenic acids (18:3),

eicosatetraenoic acids (20:5), stearidonic acids (18:4), docosahexaenoic acids (22:6), and clupa-nodonic acids (22:5). On the other hand, n6 fatty acids consisted of linoleic acids (18:2) and arachidonic acids (20:4).

The daily total intake of dietary n3 and n6 fatty acids was determined by calculating the average of the two 24-hour dietary intake records. Further, the daily intake of dietary n3 and n6 fatty acids, as well as the n6:n3 ratio, were categorized into three groups (T1, T2, and T3) based on tertiles.

## Covariates

The variables were analyzed in the present study as covariates, so as to control the possible confounding elements, including age, gender, race, sleeping disorder, education level, poverty-income ratio (PIR), body mass index (BMI), marital status, work activity, recreational activity, smoking status, drinking status, diabetes, hypertension, hypercholesterolemia and energy intake. Detailed classification and definition of the covariates were shown in S2 Table.

## Statistical analysis

In the present study, Stata 15.0 (Stata Corp, College Station, TX, USA) was used for the primary statistical analyses. Following the suggestions of NHANES, a new 12-year sample weight was recalculated. The normality of quantitative variables was tested using the Kolmogoro–Smirnov method. Then, Student's t-test was employed to compare individual characteristics between groups with and without stroke events for normally distributed continuous data, the Mann–Whitney U test for continuous data with skewed distributions, and the Chi-square test for categorical variables.

The logistic regression model was used to analyze the associations between dietary n3 and n6 fatty acids intake, as well as the n6:n3 ratio, and the risk of stroke. The reference group for this analysis was the one with the lowest dietary n3 and n6 fatty acids intake, as well as the lowest n6:n3 ratio, which corresponds to group T1. The results were reported as odds ratios (ORs) and 95% confidence intervals (CIs). To assess effect modification, a stratified analysis was employed to check whether the correlation differed by sex and age. In multivariate logistic regressions, Model 1 was adjusted for age and gender, and Model 2 was adjusted for all the variables in S1 Table. The results were statistically significant when the two-sided $p$ value was less than 0.05.

## Results

The present study included 29,459 participants, 14,631 (49.67%) males and 14,828 (50.33%) females, and the average age was 49.87 ± 17.58 years. Among the 29,459 participants, 1,121 (3.81%) individuals experienced a stroke. The baseline distribution characteristics of the individuals were demonstrated as stroke and non-stroke (Table 1). Participants exhibited several characteristics that differed from those who did not have a stroke. They tended to be older, female, smokers, obese, married or living with a partner, and drinkers. Moreover, they were more likely to have a history of hypertension, hypercholesterolemia, and diabetes. Additionally, stroke cases had a higher level of education, higher income (PIR), and engaged in less physical activity. Notably, individuals who had experienced a stroke also had lower dietary intake of n3 and n6 fatty acids compared to those without a stroke.

The results of logistic regression analysis, as shown in Table 2, were reported in terms of ORs and 95% CIs. In the crude model, individuals in the highest tertile (T3) of dietary n3 and n6 fatty acids intake exhibited a 46% and 53% reduced likelihood of experiencing a stroke event, with ORs and 95% CIs of 0.54 (0.44–0.67) and 0.47 (0.39–0.58), respectively, when

**Table 1. Characteristics of participants by stroke status, NHANES 2007–2018.**

| | Stroke | Non-stroke | *P* value |
|---|---|---|---|
| Number of subjects (%) | 1121 (3.81) | 28338 (96.19) | |
| Age (%)[1] | | | < 0.001 |
| 18–39 | 52 (4.64) | 9547 (33.69) | |
| 40–59 | 258 (23.02) | 9615 (33.93) | |
| ≥60 | 811 (72.35) | 9176 (32.38) | |
| Gender (%)[1] | | | 0.988 |
| Male | 557 (49.69) | 14074 (49.66) | |
| Female | 564 (50.31) | 14264 (50.34) | |
| Race (%)[1] | | | < 0.001 |
| Mexican American | 97 (8.65) | 4301 (15.18) | |
| Other Hispanic | 72 (6.42) | 2991 (10.55) | |
| Non-Hispanic White | 550 (49.06) | 11806 (41.66) | |
| Non-Hispanic Black | 317 (28.28) | 6012 (21.22) | |
| Other Race | 85 (7.58) | 3228 (11.39) | |
| Educational level (%)[1] | | | < 0.001 |
| <High school | 377 (33.66) | 6657 (23.51) | |
| High school | 311 (27.77) | 6469 (22.85) | |
| >High school | 432 (38.57) | 15186 (53.64) | |
| Body mass index (BMI) (%)[1] | | | < 0.001 |
| <25 kg/m$^2$ | 266 (23.99) | 8033 (28.39) | |
| 25–30 kg/m$^2$ | 349 (31.47) | 9315 (32.92) | |
| ≥30 kg/m$^2$ | 494 (44.54) | 10952 (38.70) | |
| Poverty-income ratio (%)[1] | | | < 0.001 |
| <1.00 | 266 (25.78) | 5422 (21.01) | |
| ≥ 1.00 | 766 (74.22) | 20379 (78.99) | |
| Smoking status (%)[1] | | | < 0.001 |
| Yes | 687 (61.28) | 12476 (44.04) | |
| No | 434 (38.72) | 15850 (55.96) | |
| Drinking status (%)[1] | | | < 0.001 |
| Yes | 928 (88.38) | 24414 (92.73) | |
| No | 122 (11.62) | 1915 (7.27) | |
| Work activity (%)[1] | | | < 0.001 |
| Vigorous | 150 (13.40) | 5768 (20.36) | |
| Moderate | 195 (17.43) | 6106 (21.56) | |
| Other | 774 (69.17) | 16452 (58.08) | |
| Recreational activity (%)[1] | | | < 0.001 |
| Vigorous | 58 (5.18) | 6441 (22.73) | |
| Moderate | 250 (22.32) | 7301 (25.77) | |
| Other | 812 (72.50) | 14592 (51.50) | |
| Marital Status (%)[1] | | | < 0.001 |
| Married/Living with partner | 571 (50.98) | 16828 (59.41) | |
| Widowed/Divorced/Separated | 448 (40.00) | 6149 (21.71) | |
| Never married | 101 (9.02) | 5349 (18.88) | |
| Sleeping disorder (%)[1] | | | < 0.001 |
| Yes | 494 (44.11) | 7232 (25.53) | |
| No | 626 (55.89) | 21101 (74.47) | |
| Hypertension status(%)[1] | | | < 0.001 |

*(Continued)*

**Table 1.** (Continued)

| | Stroke | Non-stroke | *P* value |
|---|---|---|---|
| Yes | 926 (83.35) | 13890 (49.96) | |
| No | 185 (16.65) | 13910 (50.04) | |
| Diabetes status(%)[1] | | | < 0.001 |
| Yes | 392 (34.97) | 3753 (13.24) | |
| No | 729 (65.03) | 24585 (86.76) | |
| Hypercholesterolemia status(%)[1] | | | < 0.001 |
| Yes | 675 (62.21) | 9791 (38.75) | |
| No | 410 (37.79) | 15476 (61.25) | |
| Total n3 fatty acids intake(g/day)[2] | 1.50 (1.04) | 1.69 (1.08) | < 0.001 |
| Total n6 fatty acids intake(g/day)[2] | 13.12 (8.16) | 15.03 (8.79) | < 0.001 |
| Total energy intake (kcal/d)[2] | 1693.79 (770.95) | 1929.63 (827.18) | < 0.001 |

[1] Chi-square test was used to compare the percentage between participants with and without stroke.

[2] Student's t-test was used to compare the mean values between participants with and without stroke

compared to those in the lowest tertile (T1). Similar findings were observed in Model 1. In Model 2, after adjusting for covariates, a higher intake of dietary n3 and n6 fatty acids remained associated with a lower risk of stroke events. The ORs with 95% CIs for individuals in Tertile 3 of stroke events were 0.67 (0.49–0.93) and 0.65 (0.45–0.95), respectively. However, the negative correlation between the n6:n3 ratio and stroke was not statistically significant in all three models.

**Table 2. Weighted ORs (95%CIs) for stroke according to tertiles of dietary intake of n3, n6 fatty acids, NHANES 2007–2018.**

| Intake cutoff | | Cases/participants[1] | Crude[2] | Model 1[2] | Model 2[2] |
|---|---|---|---|---|---|
| | | | OR (95%CI) | OR (95%CI) | OR (95%CI) |
| **Dietary n3 fatty acids intake(g/day)** | | | | | |
| Tertile 1 (low) | < 1.11 | 498/9823 | 1 (ref) | 1 (ref) | 1 (ref) |
| Tertile 2 | 1.11–1.87 | 328/9821 | 0.59 (0.46–0.75)** | 0.61 (0.47–0.79)** | 0.70 (0.51–0.95)* |
| Tertile 3(high) | ≥ 1.87 | 295/9815 | 0.54 (0.44–0.67)** | 0.57 (0.46–0.71)** | 0.67 (0.49–0.93)** |
| **Dietary n6 fatty acids intake(g/day)** | | | | | |
| Tertile 1 (low) | < 10.26 | 493/9820 | 1 (ref) | 1 (ref) | 1 (ref) |
| Tertile 2 | 10.26–16.88 | 353/9820 | 0.68 (0.54–0.87)* | 0.74 (0.58–0.95)* | 0.85 (0.61–1.18) |
| Tertile 3(high) | ≥ 16.88 | 275/9819 | 0.47 (0.39–0.58)** | 0.55 (0.44–0.68)** | 0.65 (0.45–0.95)* |
| **n6: n3 ratio** | | | | | |
| Tertile 1 (low) | < 8.08 | 395/9818 | 1 (ref) | 1 (ref) | 1 (ref) |
| Tertile 2 | 8.08–9.97 | 374/9818 | 1.07 (0.86–1.32) | 1.17 (0.94–1.46) | 1.14 (0.89–1.45) |
| Tertile 3(high) | ≥ 9.97 | 352/9818 | 0.86 (0.68–1.09) | 1.03 (0.80–1.32) | 0.99 (0.75–1.30) |

OR, odd ratio; CI, confidence interval.

[1] Cases of stroke/number of participants in tertiles.

[2] Calculated using binary logistic regression.

Model 1 adjusted for age and gender. Model 2 adjusted for age, gender, race, body mass index, marital status, poverty-income ratio, educational level, smoking status, sleeping disorder, drinking status, work activity, recreational activity, hypertension status, hypercholesterolemia status, diabetes status and energy intake.

Results were dietary-weighted.

*$p<0.05$

**$p<0.01$

**Table 3. Association between dietary n3, n6 fatty acids intake and stroke after sex stratification, NHANES 2007–2018.**

| | Male | | | Female | | |
|---|---|---|---|---|---|---|
| | Crude[1] | Model 1[1] | Model 2[1] | Crude[1] | Model 1[1] | Model 2[1] |
| **Dietary n3 fatty acids intake(g/day)** | | | | | | |
| < 1.11 | 1 (ref) | 1 (ref) | 1 (ref) | 1 (ref) | 1 (ref) | 1 (ref) |
| 1.11–1.87 | 0.53 (0.39–0.72)** | 0.54 (0.39–0.73)** | 0.63 (0.42–0.95)* | 0.64 (0.45–0.90)** | 0.66 (0.47–0.94)* | 0.75 (0.53–1.06) |
| ≥ 1.87 | 0.54 (0.38–0.76)** | 0.57 (0.41–0.81)** | 0.72 (0.44–1.18) | 0.55 (0.40–0.76)** | 0.55 (0.40–0.76)** | 0.62 (0.39–0.94)* |
| *P-interaction* | 0.997 | | | | | |
| **Dietary n6 fatty acids intake(g/day)** | | | | | | |
| < 10.26 | 1 (ref) | 1 (ref) | 1 (ref) | 1 (ref) | 1 (ref) | 1 (ref) |
| 10.26–16.88 | 0.56 (0.40–0.80)** | 0.60 (0.42–0.85)** | 0.62 (0.39–0.99)* | 0.77 (0.56–1.07) | 0.84 (0.60–1.18) | 1.04 (0.71–1.53) |
| ≥ 16.88 | 0.41 (0.30–0.57)** | 0.47 (0.34–0.65)** | 0.53 (0.32–0.85)** | 0.54 (0.39–0.74)** | 0.61 (0.44–0.84)** | 0.77 (0.45–1.32) |
| *P-interaction* | 0.852 | | | | | |
| **n6:n3 ratio** | | | | | | |
| < 8.08 | 1 (ref) | 1 (ref) | 1 (ref) | 1 (ref) | 1 (ref) | 1 (ref) |
| 8.08–9.97 | 0.79 (0.56–1.12) | 0.87 (0.61–1.24) | 0.88 (0.62–1.24) | 1.38 (1.03–1.85)* | 1.49 (1.11–2.01)** | 1.40 (0.98–1.99) |
| ≥ 9.97 | 0.63 (0.44–0.91)* | 0.72 (0.50–1.05) | 0.78 (0.54–1.13) | 1.13 (0.84–1.51) | 1.37 (1.02–1.85)* | 1.19 (0.83–1.70) |
| *P-interaction* | 0.276 | | | | | |

[1]Calculated using binary logistic regression. Model 1 adjusted for age. Model 2 adjusted for age, race, body mass index, marital status, poverty-income ratio, educational level, smoking status, sleeping disorder, drinking status, work activity, recreational activity, hypertension status, hypercholesterolemia status, diabetes status and energy intake.

A stratification analysis by sex was conducted to investigate the relationship between dietary intake of n3 and n6 fatty acids and the risk of stroke, as presented in Table 3. Among males, it was observed that the second tertile (T2) of dietary n3 fatty acids intake was inversely associated with the risk of stroke events, with an OR of 0.63 (95% CI: 0.42–0.95). Additionally, Tertile 2 and Tertile 3 of dietary n6 fatty acids intake exhibited an inverse correlation with the risk of stroke events among males, with ORs (95% CIs) of 0.62 (0.39–0.99) and 0.53 (0.32–0.85), respectively, in Model 2. Conversely, among females, it was found that only the third tertile (T3) of dietary n3 fatty acid intake had a negative association with stroke events, with an OR (95% CI) of 0.62 (0.39–0.94) in Model 2. Further, there was no statistically significant interaction observed between dietary n3 (P-interaction = 0.997) or n6 (P-interaction = 0.852) fatty acids intake and sex concerning stroke risk.

Subsequently, an age stratified analysis was conducted to assess whether the association between dietary intake of n3 and n6 fatty acids and the risk of stroke varied by age, as detailed in Table 4. Notably, there was a significant interaction observed between dietary n3 (P-interaction <0.001) and n6 (P-interaction <0.001) fatty acids and age concerning stroke events. The results revealed that the inverse relationship between dietary n3 and n6 fatty acids and stroke risk was more pronounced in the group aged 60 years or older. Within this age group, individuals in Tertile 2 and Tertile 3 of dietary n3 fatty acid intake exhibited a negative association with the risk of stroke events, with ORs (95% CIs) of 0.63 (0.46–0.86) and 0.59 (0.39–0.88), respectively. Moreover, the third tertile (T3) of dietary n6 fatty acids intake was also negatively associated with stroke events, with an OR (95% CI) of 0.61 (0.37–0.99).

## Discussion

In the present study, the associations between dietary n3 and n6 fatty acids intake, as well as the n6:n3 ratio, and the risk of stroke were explored within the American population. Data

**Table 4. Association between dietary n3, n6 fatty acids intake and stroke after age stratification, NHANES 2007–2018.**

| | 18–39 years old | | | 40–59 years old | | | ≥60 years old | | |
|---|---|---|---|---|---|---|---|---|---|
| | Crude[1] | Model 1[1] | Model 2[1] | Crude[1] | Model 1[1] | Model 2[1] | Crude[1] | Model 1[1] | Model 2[1] |
| **Dietary n3 fatty acids intake(g/day)** | | | | | | | | | |
| < 1.11 | 1 (ref) | 1 (ref) | 1 (ref) | 1 (ref) | 1 (ref) | 1 (ref) | 1 (ref) | 1 (ref) | 1 (ref) |
| 1.11–1.87 | 0.63 (0.22–1.79) | 0.65 (0.23–1.86) | 0.99 (0.35–2.82) | 0.59 (0.37–0.94)* | 0.60 (0.37–0.96)* | 0.87 (0.48–1.58) | 0.61 (0.47–0.80)** | 0.61 (0.47–0.80)** | 0.63 (0.46–0.86)** |
| ≥ 1.87 | 0.80 (0.27–2.38) | 0.86 (0.30–2.49) | 0.52 (0.09–2.86) | 0.71 (0.47–1.07) | 0.74 (0.48–1.14) | 0.95 (0.46–1.96) | 0.48 (0.37–0.63)** | 0.48 (0.37–0.62)** | 0.59 (0.39–0.88)* |
| P-interaction | <0.001 | | | | | | | | |
| **Dietary n6 fatty acids intake(g/day)** | | | | | | | | | |
| < 10.26 | 1 (ref) | 1 (ref) | 1 (ref) | 1 (ref) | 1 (ref) | 1 (ref) | 1 (ref) | 1 (ref) | 1 (ref) |
| 10.26–16.88 | 1.00 (0.33–2.97) | 1.02 (0.34–3.02) | 1.98 (0.56–7.03) | 0.78 (0.50–1.22) | 0.79 (0.50–1.23) | 1.14 (0.64–2.01) | 0.71 (0.54–0.92)** | 0.70 (0.54–0.92)* | 0.74 (0.52–1.04) |
| ≥ 16.88 | 0.72 (0.27–1.90) | 0.77 (0.29–2.05) | 1.98 (0.50–7.84) | 0.61 (0.40–0.93)* | 0.63 (0.40–0.99)* | 0.75 (0.36–1.57) | 0.51 (0.39–0.65)** | 0.50 (0.38–0.65)** | 0.61 (0.37–0.99)* |
| P-interaction | <0.001 | | | | | | | | |
| **n6:n3 ratio** | | | | | | | | | |
| < 8.08 | 1 (ref) | 1 (ref) | 1 (ref) | 1 (ref) | 1 (ref) | 1 (ref) | 1 (ref) | 1 (ref) | 1 (ref) |
| 8.08–9.97 | 1.16 (0.33–4.11) | 1.16 (0.32–4.17) | 2.28 (0.66–7.93) | 1.43 (0.91–2.26) | 1.45 (0.91–2.31) | 1.34 (0.80–2.42) | 1.06 (0.79–1.41) | 1.06 (0.79–1.41) | 1.01 (0.73–1.40) |
| ≥ 9.97 | 1.02 (0.30–3.53) | 1.04 (0.30–3.53) | 1.93 (0.69–5.39) | 0.84 (0.50–1.40) | 0.85 (0.51–1.41) | 0.83 (0.46–1.49) | 1.13 (0.84–1.51) | 1.13 (0.84–1.53) | 1.07 (0.77–1.47) |
| P-interaction | 0.987 | | | | | | | | |

[1]Calculated using binary logistic regression. Model 1 adjusted for gender. Model 2 adjusted for gender, race, body mass index, marital status, poverty-income ratio, educational level, smoking status, sleeping disorder, drinking status, work activity, recreational activity, hypertension status, hypercholesterolemia status, diabetes status and energy intake.

from 29,459 individuals in NHANES were included (from 2007 to 2018). The present results indicated that the intake of n3 and n6 fatty acids was negatively associated with the risk of stroke, but n6:n3 ratio was not associated with a stroke event. Results from stratified analysis demonstrated that the intake of dietary n3 fatty acids had an inverse correlation with stroke risk in both males and females. However, the correlation between dietary n6 fatty acids and stroke was observed only in males. Further, the negative relationship between dietary n3 and n6 fatty acids intake and the risk of stroke was notably pronounced among participants aged ≥60.

The significant inverse relationship between dietary n3 fatty acids and stroke observed in the present study aligned with findings from several prior studies. For instance, a prospective study conducted among women in the Nurses' Health Study cohort also reported that women in the highest quintile of long-chain n3 fatty acids intake had a decreased risk of stroke [24]. This inverse association between long-chain n3 fatty acids and stroke event in women was also observed in a large population-based study in the Netherlands and a cohort study in Sweden [26, 27]. During 12 years of follow-up, Bergkvist et al. also observed that the consumption of n3 fatty acids had a protective role on the risk of stroke [28]. In another prospective population-based cohort study involving 39,948 Swedish men, EPA-DHA intake was confirmed to be associated with hemorrhagic stroke but not ischemic stroke [30]. Nevertheless, three studies showed no significant relationship between dietary n3 fatty acids and stroke [23, 25, 29]. In

contrast to the present findings, the Swedish Mammography Cohort study, comprising 34,670 women, did not reveal an association between dietary n6 fatty acids and stroke incidence.

The beneficial impact of dietary n3 and n6 fatty acid intake on stroke risk may be attributed, in part, to its antihypertensive properties [37], given that hypertension is a significant risk factor for stroke [38]. However, in the present study, the inverse correlation between dietary n3 and n6 fatty acids and stroke risk remained statistically significant even after adjusting for hypertension. Besides blood pressure-lowering effects, several potential mechanisms have been proposed. Firstly, a high intake of n3 fatty acids can mitigate platelet aggregation and vasoconstriction by reducing the production of thromboxane A2 in platelets and enhancing the synthesis of the vasodilator prostaglandin I3 [39–41]. Secondly, increased n3 fatty acid consumption has been associated with a reduction in the intrinsic clotting pathway and a decrease in blood viscosity [28, 42]. Further, n3 and n6 fatty acids, functioning as signaling molecules, can activate peroxisome proliferation-activated receptors, thereby regulating processes such as oxidative stress, inflammatory response, lipogenesis, lipid and glucose metabolism. This multifaceted effect may contribute to a reduced risk of stroke development [43].

In the results of age stratification, it was observed that the consumption of dietary n3 and n6 fatty acids exhibited an inverse association with stroke risk specifically within the age group of 60 years and older. Additionally, an interaction effect between dietary n3 and n6 fatty acids and age on stroke risk was detected. The mechanism underlying this interaction can be elucidated by considering that age plays a pivotal role in the occurrence of stroke, with elderly individuals being at a heightened risk for stroke [44]. In the present study, the data demonstrate that the prevalence of stroke in the $\geq$60 age group was significantly higher than that in the other two groups. In addition, older individuals are more prone to reducing their fat intake due to issues related to digestion and the presence of chronic conditions such as obesity, which can lead to a decreased intake of n3 and n6 fatty acids.

This study had many novelty. Firstly, it incorporated a large, nationally representative sample of adults, thereby enhancing the statistical power and reliability of the findings. Secondly, it assessed the relationship between dietary n3 and n6 fatty acid intake and stroke, stratified by both sex and age. Thirdly, the study encompassed the inclusion and adjustment for established potential stroke risk factors, strengthening the robustness of the results.

However, several limitations should be taken into account. Firstly, the data revealed that participants who experienced strokes had lower dietary n3 and n6 fatty acids intake, while those without strokes had higher intake of these fatty acids. Consequently, the study was unable to establish a causal link between cause and effect. Secondly, nutrient intake was computed based on memory-based interviews, and although the 24-hour dietary review method was thoroughly described and validated, daily nutrient intake might not accurately reflect long-term status. Thirdly, stroke outcomes were collected through recall questionnaires, which could introduce recall bias, misdiagnosis, and potential selection bias during subject exclusion. Fourthly, the NHANES database did not differentiate between ischemic and hemorrhagic strokes, increasing the heterogeneity of stroke outcomes. Lastly, it remained possible that unmeasured confounding factors could have influenced the observed associations.

## Conclusion

The present findings suggested that there might be an inverse association between dietary intake of n3 and n6 fatty acids and the risk of stroke among American adults. This indicated that increasing dietary n3 and n6 fatty acids intake could potentially serve as a preventive measure for reducing the risk of stroke.

## Supporting information

**S1 Table. Detailed process of subjects exclusion.**
(DOCX)

**S2 Table. The classifications of covariates.**
(DOCX)

**S1 Data.**
(RAR)

## Author Contributions

**Data curation:** Mingya Luan, Kun Liang.

**Formal analysis:** Mingya Luan, Bo Li.

**Methodology:** Jia Wang, Kewei Liu.

**Software:** Jia Wang.

**Supervision:** Kewei Liu.

**Writing – original draft:** Mingya Luan, Kun Liang.

**Writing – review & editing:** Jia Wang.

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
