## [Decision Letter · Decision Letter 0]

22 Aug 2023

PONE-D-23-13938Association of dietary n3 and n6 fatty acids intake with stroke in US adults: a cross-sectional study of NHANES 2007-2018PLOS ONE

Dear Dr. Liu,

Thank you for submitting your manuscript to PLOS ONE. After careful consideration, we feel that it has merit but does not fully meet PLOS ONE’s publication criteria as it currently stands. Therefore, we invite you to submit a revised version of the manuscript that addresses the points raised during the review process.

We look forward to receiving your revised manuscript.

Kind regards,

Xinlin Zhang

Academic Editor

PLOS ONE

Journal Requirements:

- https://www.mdpi.com/2072-6643/11/6/1232/html

In your revision ensure you cite all your sources (including your own works), and quote or rephrase any duplicated text outside the methods section. Further consideration is dependent on these concerns being addressed.

"NO authors have competing interests"

Additional Editor Comments:

Several important issues have been raised by the 2 reviewers, please address all the issues and provide rational, where appropriate. Languages should also be polished. Please understand that this revision does not guarantee acceptance.

Reviewers' comments:

Reviewer's Responses to Questions

**Comments to the Author**

1. Is the manuscript technically sound, and do the data support the conclusions?

Reviewer #1: Yes

Reviewer #2: Partly

2. Has the statistical analysis been performed appropriately and rigorously? 

Reviewer #1: Yes

Reviewer #2: No

3. Have the authors made all data underlying the findings in their manuscript fully available?

Reviewer #1: Yes

Reviewer #2: Yes

4. Is the manuscript presented in an intelligible fashion and written in standard English?

Reviewer #1: No

Reviewer #2: Yes

5. Review Comments to the Author

Reviewer #1: The authors used the data from the 2007–2018 National Health and Nutrition Examination Survey (NHANES) to study the association between dietary n3, n6 fatty acids intake and stroke in a sample of 29,459 adults. They found that found that dietary n3 and n6 fatty acids intake were inversely associated with stroke in US adults. The novelty of the study is limited.

1. Abstract: The statement “The association between dietary n3, n6 fatty acids intake and the risk of stroke has not been investigated in the general population.” seems problematic. Supported by literature?

2. Introduction: The sentence “Given the negative impact of stroke on individuals and society, it is important to find effective non-drug interventions to reduce the incidence of stroke [5].” appears meaningless, which should be deleted.

3. The intake of fatty acids was not fully defined. Daily intake? The investigators seemed able to estimate the intake on a cross-sectional basis only, while the impact of fatty acids must be a long-term one. The authors are unable to explain why individuals at some certain age failed to present the inverse associations.

4. The authors are supposed to do stratified analyses, e.g. ischemic stroke and cerebral hemorrhage, because stroke per se is highly heterogeneous.

5. Language：the manuscript is in need of English language editing. There appear many errors in English grammar, syntax, and word usage including, but are not limited to, lack of subject-verb agreement (singular vs. plural), inconsistent tense usage, improper word forms (noun vs. verb vs. adjective vs. gerund). Accordingly, the reviewer suggests that the authors have the text professionally edited by an English improvement service before the manuscript can be reconsidered for publication in the journal. For example, grammatical error is seen in the sentences “Stroke is also one of leading causes of long-term disability worldwide, leading a high health care expenditures and low quality of life for those survive from it[3].” “Herein, we use a large sample in the National Health and Nutrition Examination Survey (NHANES) from 2007-2018 to calculated the associations of dietary n3 and n6 fatty acids intake and the risk of stroke.”

Reviewer #2: In this study, the association between dietary n3 and n6 fatty acid intake and the risk of stroke was investigated. The authors concluded that dietary intake of n3 and n6 fatty acids might be inversely associated with the risk of stroke in US adults. However, there are some concerns.

1. As mentioned by the authors in the limitations section, the primary weakness of this study is its cross-sectional design, which precludes establishing causation. While the authors discuss potential causal relationships between dietary n3 and n6 fatty acid intake and stroke risk, this discussion might not be appropriate. The discussion should be grounded in the observed results, specifically that participants with stroke had lower dietary n3 and n6 fatty acid intake, whereas those without stroke had higher intake of these fatty acids. Therefore, the authors are likely unable to draw causal inferences regarding cause and effect.

2. Another notable limitation is the definition of the outcome, stroke, based on the Medical Conditions Questionnaire. This raises questions about its validity. Given the heterogeneous nature of stroke, standardized clinical criteria should ideally be used for diagnosis. The potential for misdiagnosis or recall bias due to this definition should be acknowledged. It is advisable to provide references that support the acceptability of this questionnaire-based definition.

3. Were interactions examined in subgroup analyses by the authors? Were interactions examined in subgroup analyses by the authors? Any statistical differences observed within subgroups should be discussed in the context of the results.

4. Could the exclusion of certain participants introduce selection bias? This possibility should be considered and addressed in the limitations section.

5. The study covers a 12-year span of data accumulation. Has there been any notable change in dietary n3 and n6 fatty acid intake or in the incidence of stroke during this period? Exploring such potential changes could provide valuable context to the study's findings.

6. PLOS authors have the option to publish the peer review history of their article (what does this mean?). If published, this will include your full peer review and any attached files.

Reviewer #1: **Yes: **Hongliang Zhang

Reviewer #2: No

---

## [Author Response · Author response to Decision Letter 0]

4 Sep 2023

Comments to the Author

1. Is the manuscript technically sound, and do the data support the conclusions?

Reviewer #1: Yes

Reviewer #2: Partly

In our manuscript, we described a technically sound piece of scientific research and the data could support the conclusions.

2.Has the statistical analysis been performed appropriately and rigorously?

Reviewer #1: Yes

Reviewer #2: No

We have strictly followed the data processing methods of the NHANES database and applied the appropriate analysis methods and models, the same methods have been used in previous published articles.

3.Have the authors made all data underlying the findings in their manuscript fully available?

Reviewer #1: Yes

Reviewer #2: Yes

We have made all data underlying the findings in our manuscript fully available.

4.Is the manuscript presented in an intelligible fashion and written in standard English?

Reviewer #1: No

Reviewer #2: Yes

In the revised manuscript, we have carefully corrected the errors.

5.Review Comments to the Author

Reviewer #1: The authors used the data from the 2007–2018 National Health and Nutrition Examination Survey (NHANES) to study the association between dietary n3, n6 fatty acids intake and stroke in a sample of 29,459 adults. They found that found that dietary n3 and n6 fatty acids intake were inversely associated with stroke in US adults. The novelty of the study is limited.

1.Abstract: The statement “The association between dietary n3, n6 fatty acids intake and the risk of stroke has not been investigated in the general population.” seems problematic. Supported by literature?

Reply: Thank you very much for your questions. It is imprudent for us to say so, and we have corrected it in the revised manuscript. Please seen line 24-26 on page 2 for detail.

2. Introduction: The sentence “Given the negative impact of stroke on individuals and society, it is important to find effective non-drug interventions to reduce the incidence of stroke [5].” appears meaningless, which should be deleted.

Reply: Thank you very much for your comments. We have deleted this sentence in the revised manuscript. Please seen line 58 on page 2 for detail.

3. The intake of fatty acids was not fully defined. Daily intake? The investigators seemed able to estimate the intake on a cross-sectional basis only, while the impact of fatty acids must be a long-term one. The authors are unable to explain why individuals at some certain age failed to present the inverse associations.

Reply: Thank you very much for your in-depth questions. It is daily dietary intake of n3, n6 fatty acids calculated in our study, and we have made it clear in the revised manuscript. Please seen line 109-110 on page 5 for detail. The NHANES database uses a method of 24-hour dietary review to assess the nutritional health status of residents, and long-term nutrient monitoring data were not available in this database. This is a major shortcoming of cross-sectional studies, and we have summarized it in the limitation section of the discussion. Please seen line 239-242 on page 24 for detail. In our study, we fail to find a significant negative association between dietary n3, n6 fatty acids intake and the risk of stroke in people aged 19-39 and 40-59, possibly because there are fewer stroke cases in the two age groups, thus weakening the association between dietary n3, n6 fatty acids and stroke. Moreover, we have explored the interaction of age in the relationship between dietary n3, n6 fatty acids and stroke in the revised manuscript. We found that there was an interaction between dietary n3 (P-interaction ＜0.001), n6 (P-interaction ＜0.001) fatty acids and age on stroke event. Please seen line 171-173 on page 16 and line 180-181 on page 19 for detail.

4. The authors are supposed to do stratified analyses, e.g. ischemic stroke and cerebral hemorrhage, because stroke per se is highly heterogeneous.

Reply: Thank you very much for your advice. Unfortunately, the NHANES database does not distinguish between types of stroke, so our study can only suggest an association between dietary n3, n6 fatty acids and stroke event, and does not clarify whether the stroke was haemorrhagic or ischemic.

5. Language：the manuscript is in need of English language editing. There appear many errors in English grammar, syntax, and word usage including, but are not limited to, lack of subject-verb agreement (singular vs. plural), inconsistent tense usage, improper word forms (noun vs. verb vs. adjective vs. gerund). Accordingly, the reviewer suggests that the authors have the text professionally edited by an English improvement service before the manuscript can be reconsidered for publication in the journal. For example, grammatical error is seen in the sentences “Stroke is also one of leading causes of long-term disability worldwide, leading a high health care expenditures and low quality of life for those survive from it[3].” “Herein, we use a large sample in the National Health and Nutrition Examination Survey (NHANES) from 2007-2018 to calculated the associations of dietary n3 and n6 fatty acids intake and the risk of stroke.”

Reply: Thank you very much for your advice. We have corrected the errors reduced the repetition rate in the revised manuscript. Please review it again.

Reviewer #2: In this study, the association between dietary n3 and n6 fatty acid intake and the risk of stroke was investigated. The authors concluded that dietary intake of n3 and n6 fatty acids might be inversely associated with the risk of stroke in US adults. However, there are some concerns.

1. As mentioned by the authors in the limitations section, the primary weakness of this study is its cross-sectional design, which precludes establishing causation. While the authors discuss potential causal relationships between dietary n3 and n6 fatty acid intake and stroke risk, this discussion might not be appropriate. The discussion should be grounded in the observed results, specifically that participants with stroke had lower dietary n3 and n6 fatty acid intake, whereas those without stroke had higher intake of these fatty acids. Therefore, the authors are likely unable to draw causal inferences regarding cause and effect.

Reply: Thank you very much for your in-depth questions. We have revised this limitation in the discussion section. Please seen line 236-239 on page 24 for detail.

2. Another notable limitation is the definition of the outcome, stroke, based on the Medical Conditions Questionnaire. This raises questions about its validity. Given the heterogeneous nature of stroke, standardized clinical criteria should ideally be used for diagnosis. The potential for misdiagnosis or recall bias due to this definition should be acknowledged. It is advisable to provide references that support the acceptability of this questionnaire-based definition.

Reply: Thanks for the advice.It is a pity that the NHANEA database does not include the stroke data from standardized clinical criteria. The measures of self-reported stroke by questionnaire are reliable and have been employed in many previous articles. In the revised manuscript, we have provided references to support this point. Please seen line 97-98 on page 5 for detail. Compared with the clinical standard diagnosis results, the stroke diagnosis obtained from the questionnaire may indeed lead to misdiagnosis and recall bias. We have summarized it in the limitation section. Please seen line 242-244 on page 24 for detail.

3. Were interactions examined in subgroup analyses by the authors? Any statistical differences observed within subgroups should be discussed in the context of the results.

Reply: Thanks for your suggestion. We have explored the interactions in subgroup analyses, and the results were showed in Table 3 and Table 4. The results indicated that subgroup analyses by sex showed no statistically significant interactions, while subgroup analyses by age showed a significant interactions. We have summarized these in the results. Please seen line 171-173 on page 16 and line 180-181 on page 19 for detail.

4. Could the exclusion of certain participants introduce selection bias? This possibility should be considered and addressed in the limitations section.

Reply: Thanks for your suggestion. We have added this in the limitation section.Please seen line 243-244 on page 24 for detail.

5. The study covers a 12-year span of data accumulation. Has there been any notable change in dietary n3 and n6 fatty acid intake or in the incidence of stroke during this period? Exploring such potential changes could provide valuable context to the study's findings.

Reply: Thanks for very much for your suggestion. We have analyzed the data from the six cycles, and the results are shown in the file of response to reviewers (Table 1 and Figure1-2). From the results, we can found that dietary n3, n6 fatty acids intake shows an overall upward trend from 2007 to 2018, but the prevalence of stroke fluctuated. Therefore, we do not discuss this result in the manuscript.

---

## [Decision Letter · Decision Letter 1]

6 Oct 2023

PONE-D-23-13938R1Association of dietary n3, n6 fatty acids intake with stroke in US adults: a cross-sectional study of NHANES 2007-2018PLOS ONE

Dear Dr. Liu,

Thank you for submitting your manuscript to PLOS ONE. After careful consideration, we feel that it has merit but does not fully meet PLOS ONE’s publication criteria as it currently stands. Therefore, we invite you to submit a revised version of the manuscript that addresses the points raised during the review process.

We look forward to receiving your revised manuscript.

Kind regards,

Xinlin Zhang

Academic Editor

PLOS ONE

Journal Requirements:

Additional Editor Comments (if provided):

The manuscript requires thorough language editing, preferably by a native speaker, to ensure the best quality. Additionally, the Discussion section needs to be revised to accurately reflect the association between fatty acid intake and stroke risk, but not a causal relationship. Furthermore, a paragraph should be added to describe the differences and mechanisms observed in stroke risk across different age groups. It is important to include a limitation section acknowledging the heterogeneity of stroke outcomes and the inability to stratify between ischemic and hemorrhagic strokes.

Reviewers' comments:

Reviewer's Responses to Questions

**Comments to the Author**

1. If the authors have adequately addressed your comments raised in a previous round of review and you feel that this manuscript is now acceptable for publication, you may indicate that here to bypass the “Comments to the Author” section, enter your conflict of interest statement in the “Confidential to Editor” section, and submit your "Accept" recommendation.

Reviewer #1: (No Response)

Reviewer #2: All comments have been addressed

2. Is the manuscript technically sound, and do the data support the conclusions?

Reviewer #1: (No Response)

Reviewer #2: Yes

3. Has the statistical analysis been performed appropriately and rigorously? 

Reviewer #1: (No Response)

Reviewer #2: Yes

4. Have the authors made all data underlying the findings in their manuscript fully available?

Reviewer #1: (No Response)

Reviewer #2: Yes

5. Is the manuscript presented in an intelligible fashion and written in standard English?

Reviewer #1: (No Response)

Reviewer #2: Yes

6. Review Comments to the Author

Reviewer #1: This is a revised version of the previously submitted study. The reviewers’ concerns cannot be addressed. As such, the reviewer would suggest against its publication and further consideration. The bottleneck is that the NHANES database does not distinguish between types of stroke, so their analysis can only suggest an association between dietary n3, n6 fatty acids and stroke event, and does not clarify whether the stroke was hemorrhagic or ischemic.

Reviewer #2: (No Response)

7. PLOS authors have the option to publish the peer review history of their article (what does this mean?). If published, this will include your full peer review and any attached files.

Reviewer #1: **Yes: **Hongliang Zhang

Reviewer #2: No

---

## [Author Response · Author response to Decision Letter 1]

14 Oct 2023

Response to Reviewers

Manuscript title: Association of dietary n3, n6 fatty acids intake with stroke in US adults: a cross-sectional study of NHANES 2007-2018

Manuscript ID: PONE-D-23-13938

Journal Requirements:

Reply: We have reviewed the list of reference and we did not cite retracted paper in my manuscript. But in this progress, we found same mistakes and we have corrected them in the revised manuscript. Please seen Line 297, Line 300-301, Line 338 in Page 26-27.

The manuscript requires thorough language editing, preferably by a native speaker, to ensure the best quality. Additionally, the Discussion section needs to be revised to accurately reflect the association between fatty acid intake and stroke risk, but not a causal relationship. Furthermore, a paragraph should be added to describe the differences and mechanisms observed in stroke risk across different age groups. It is important to include a limitation section acknowledging the heterogeneity of stroke outcomes and the inability to stratify between ischemic and hemorrhagic strokes.

Reply: The language of the manuscript was edited throughout, and we also revised the section of discussion, please seen Line 213-219 in Page 22. In the revised manuscript, a paragraph has been added to describe the differences and mechanisms observed in stroke risk across different age groups. Please seen Line 250-260 in Page 24. We have added the disadvantage of not being able to stratifying ischemic and hemorrhagic strokes, Please seen Line 275-277 in Page 25.

---

## [Editor Report · Decision Letter 2]

23 Oct 2023

Association between the intake of dietary n3 and n6 fatty acids and stroke in US adults: a cross-sectional study of NHANES 2007-2018

PONE-D-23-13938R2

Dear Dr. Liu,

We’re pleased to inform you that your manuscript has been judged scientifically suitable for publication and will be formally accepted for publication once it meets all outstanding technical requirements.

Kind regards,

Xinlin Zhang

Academic Editor

PLOS ONE
---

## [Editor Report · Acceptance letter]

7 Nov 2023

PONE-D-23-13938R2 

Association between the intake of dietary n3 and n6 fatty acids and stroke in US adults: a cross-sectional study of NHANES 2007-2018 

Dear Dr. Liu:

I'm pleased to inform you that your manuscript has been deemed suitable for publication in PLOS ONE. Congratulations! Your manuscript is now with our production department. 

Kind regards, 

on behalf of

Professor Xinlin Zhang 

Academic Editor

PLOS ONE